# Safety of Adenosine for the Treatment of Supraventricular Tachycardia in Hospitalized Patients with COVID-19 Pneumonia

**DOI:** 10.3390/jcm12030969

**Published:** 2023-01-27

**Authors:** Tal Zivan, Ramon L. Ruiz, Alexandre Martinez, Behzad B. Pavri

**Affiliations:** Department of Medicine, Thomas Jefferson University Hospital, 111 S 11th St., Philadelphia, PA 19107, USA

**Keywords:** COVID-19, arrhythmia, adenosine, SVT, safety, escalation

## Abstract

Coronavirus disease 2019 (COVID-19) is associated with pulmonary involvement and cardiac arrhythmias, including supraventricular tachycardia (SVT). Adenosine is commonly used to treat SVT and is generally safe, but is rarely associated with bronchospasm. There are no data regarding the safety of adenosine use in patients with COVID-19 pneumonia and physicians may hesitate to use it in such patients. We surveyed resident physicians and cardiology attendings regarding their level of comfort in administering adenosine to hospitalized COVID-19 patients. We compared a study group of 42 COVID-19 hospitalized patients who received adenosine for SVT to a matched (for age, sex, and co-morbidities) control group of 42 non-COVID-19 hospitalized patients during the same period, all of whom received IV adenosine for SVT. Escalation of care following intravenous adenosine administration was defined as increased/new pressor requirement, need for higher O_2_ flow rates, need for endotracheal intubation, new nebulizer therapy, or transfer to intensive care unit within 2 h of adenosine administration. Survey results showed that 82% (59/72) of residents and 62% (16/26) of cardiologists expressed hesitation/significant concerns regarding administering adenosine in hospitalized COVID-19 patients. Adenosine use was associated with escalation of care in 47.6% (20/42) COVID-19 as compared to 50% (21/42) non-COVID-19 patients (odds ratio 0.95, 95% CI 0.45–2.01, *p* = NS). Escalation of care was more likely in patients who were on higher FiO_2_, on prior nebulizer therapy, required supplemental oxygen, or were already on a ventilator. In conclusion, we identified significant hesitation among physicians regarding the use of adenosine for SVT in hospitalized COVID-19 patients. In this study, there was no evidence of increased harm from administering adenosine to patients with SVT and COVID-19. This finding needs to be confirmed in larger studies. Based on the current evidence, adenosine for treatment of SVT in this setting should not be avoided. Key Points: Question: Given the known bronchospastic effects of adenosine, is the use of adenosine safe for treatment of supraventricular tachycardia in hospitalized patients with COVID-19? Findings: A survey of residents and cardiology attending identified that a majority expressed some level of apprehension in using adenosine for SVT in COVID-19 patients. In our matched cohort study, we found adenosine use to be comparably safe in COVID-19 and non-COVID-19 hospitalized patients. Meaning: Based on current evidence, adenosine for treatment of SVT in this setting should not be avoided.

## 1. Introduction

The COVID-19 pandemic plunged medicine into chaos and uncertainty in many ways. This disease is associated with a well-described inflammatory course resulting in multi-organ consequences, including cardiopulmonary complications. Notably, one of the many cardiac complications associated with severe acute respiratory syndrome coronavirus 2 (SARS-CoV-2) is arrhythmia, with early data from Wuhan, China, showing 44% (*n* = 36) of ICU level patients exhibiting some form of arrhythmia, other larger reviews quoting a 6–17% prevalence [1], and an Italian multicenter study specifically quoting a supraventricular tachycardia (SVT) rate of 1.2% (*n* = 414) [2]. Arrhythmias are believed to be a manifestation of multiple factors including systemic inflammation, hypoxia, adrenergic state, viral myocardial damage, downregulation of angiotensin converting enzyme 2 (ACE2) pathways, volume shifts, metabolic derangements, and potential aggravation of chronic conduction disturbances [3,4].

Adenosine is a naturally occurring organic nucleoside molecule with a well-known and widely used role for SVT termination via cascade modulation of adenylate cyclase, cAMP, and potassium/calcium currents causing transient atrioventricular nodal block. However, adenosine also carries a relative contraindication in patients with asthma [5] and cautionary use in patients with COPD secondary to mast cell-mediated pulmonary bronchospasm [6], although the risk for bronchospasm is reported to be <1% [7]. Given the association of adenosine with potential pulmonary compromise and the relative novelty of SARS-CoV-2, many clinicians may avoid or hesitate to use this potent anti-arrhythmic for the treatment of SVT in such patients.

There are few data on the safety of adenosine use in patients with COVID-19. A review article analyzing arrhythmia and COVID-19 states that adenosine can be used for treatment of SVT, but recognized that more data are needed for validation [8]. Other review articles have mentioned caution with the use of beta blockers and amiodarone in the setting of COVID-19 but not with the use of adenosine [9]. On the other hand, one review suggested that the use of adenosine and its derivatives may impede SARS-CoV-2 viral entry, ARDS progression and thrombosis in the setting of COVID-19 infection, with one study from Italy showing therapeutic benefit from inhaled adenosine in the setting of COVID-19 infection [10,11]. There are many reports that focus on malignant arrhythmias, atrial fibrillation, and heart block, but there are limited data with specific attention to SVT and the safety of adenosine use [12,13].

## 2. Materials and Methods

### 2.1. Survey

We surveyed resident physicians in the fields of internal medicine, family medicine, and general surgery as well as attending cardiology physicians regarding their level of comfort in administering adenosine to patients with COVID-19 pneumonia. The survey was sent to residents from Internal Medicine, Family Medicine and General Surgery via email with an attached web link. The emails were sent out to the residents by the chief residents of each specialty rather than directly by the study investigators. Reminder emails were sent out once each week for a total of 3 weeks. The survey was also sent to all attendings in the division of cardiology as an email sent directly by the senior author (BP), with 2 reminder emails once a week for 2 weeks. The survey questions are shown in Figure 1. The same survey was also sent to the cardiology attendings, but without the last two questions.

### 2.2. Retrospective Review

We searched the electronic health record that included all five hospitals in our health system for all patients admitted with the diagnosis of COVID-19 who received adenosine for the treatment of SVT between the dates of February 01, 2020 and 1 February 2021. We compared a study group which consisted of COVID-19 hospitalized patients (*n* = 42) who received adenosine for SVT with a control group. The control group consisted of an equal number of non-COVID-19 hospitalized patients (*n* = 42) during the same period who received IV adenosine for SVT. The groups were matched for age, sex and comorbidities. Both the study group and control group had 42 patients with a mean age of 66 years old, and a gender split of 32 males and 10 females each. Additionally, the groups were matched for comorbidities including history of pulmonary disease, history of arrhythmia and history of conduction system disease. We obtained relevant medical history, including history of pulmonary disease, history of arrhythmia, and prior use of anti-arrhythmic therapy or rate controlling medications. The highest dosage of adenosine given as well as its effects on cardiac rhythm were documented. Each chart was reviewed for possible escalation of care related to adenosine and any resulting changes in management. Escalation of care was defined as any one of the following within two hours of adenosine administration: increased or new pressor requirement, need for higher O_2_ flow, endotracheal intubation, new bronchodilator nebulizer therapy, or transfer to an intensive care unit. We documented these variables before and after adenosine administration. The same electronic health record was searched (during the same time period) for age-/gender-matched hospitalized patients who received IV adenosine for SVT but who did not have an associated diagnosis of COVID-19. The exact same clinical and outcomes parameters were collected.

### 2.3. Statistical Analysis

Data were analyzed separately in both groups, and also for the entire population (both study and the control groups) based on whether or not adenosine use was associated with escalation of care as defined in our study. Continuous variables were compared between the two groups using the two-tailed Student’s t test assuming equal variances. Dichotomous variables were compared using the Fisher’s exact test. When groups had <5 data points, the Yates correction was applied. *p* values < 0.05 were considered to be statistically significant.

## 3. Results

### 3.1. Survey

The survey was sent to a total of 181 residents from Internal Medicine (109), Family Medicine (30) and General Surgery (42) via email. The response rate from residents was 39.8% (72/181). The survey was also sent to all 49 attendings in the division of cardiology and the attending response rate was 53.1% (26/49).

The survey results showed that 82% (59/72) of responding residents and 62% (16/26) of responding cardiology attendings had either some hesitation or significant concerns about administering adenosine in hospitalized COVID-19 patients. The level of hesitation was significantly greater among residents as compared to attending (*p* = 0.0353). Only 27.3% of the responding residents and 38.5% of responding cardiology attendings had no concerns about administering adenosine in COVID-19 pneumonia patients for the treatment of SVT. Survey details are shown in Figure 2.

### 3.2. Baseline Characteristics of the Study and Control Groups

Table 1 shows the clinical variables of the total study population and the two groups. There were no statistically significant differences in the study and control cohorts (except for a higher incidence of prior arrhythmia and greater need for supplemental O_2_ in the control group).

### 3.3. Study Group

Table 2 shows the baseline clinical characteristics in the 42 patients in the entire study group, and separately in patients who did not and did require escalation of care. Adenosine use for SVT was associated with escalation of care within 2 h in 47.6% (20/42) of patients. Escalation of care was significantly more likely to occur in COVID-19 patients who required prior use of nebulizers as compared to those who did not (11/20 vs. 8/22, *p* = 0.0045).

### 3.4. Control Group

Table 3 shows the baseline clinical characteristics in the 42 matched patients in the entire control group, and separately in patients who did not and did require escalation of care. Adenosine use for SVT was associated with escalation of care within 2 h in 50% (21/42) of patients. Escalation of care was significantly more likely to occur in non-COVID-19 patients who required higher pre-adenosine FIO_2_ (66.9 vs. 41.7, *p* = 0.0284), who were on supplemental oxygen at the time of adenosine administration (20/21 vs. 10/21, *p* = 0.0006), and had received previous nebulizer therapy (13/21 vs. 6/21, *p* = 0.0495) as compared to patients who did not require escalation of care.

The odds ratio of risk of escalation for COVID-19 vs. non-COVID-19 patients was 0.95, 95% CI 0.45–2.01, *p* = NS).

Among the 41 patients (from both groups) who were defined as needing escalation of care, requiring more nebulizer therapy after adenosine administration was significantly more common in the COVID-19 group as compared to the non-COVID-19 group (12/20 vs. 5/21, *p* = 0.0187); all other parameters of escalation of care (escalating pressor requirement, escalating O_2_ flow, requiring intubation, requiring nebulizers, or requiring higher transfer to an intensive care unit within two hours of adenosine administration) were comparable in the study group vs. the control group (all *p* = NS), as shown in Table 4.

## 4. Discussion

Our survey results indicate that there is apprehension/concern regarding adenosine use in COVID-19 patients among 82% of residents and 62% of cardiology attendings. According to the survey, these concerns were significantly more common amongst house staff compared to attendings. This is important since the clinical decision to administer adenosine for arrhythmia often calls for quick action. Similarly, the higher incidence of supplemental O_2_ in the control group was not associated with any differences in pre-adenosine FiO_2_ or O_2_ flow, need for nebulizers, or being on a ventilator. Overall, our data shows that there is no significant difference in the risk of requiring escalation of care after adenosine use for SVT in COVID-19 patients as compared to a matched cohort of hospitalized non-COVID-19 patients. Among patients who met our definition of escalation of care, the need for nebulizer therapy was significantly more common in the COVID-19 group as compared to the control group, suggesting that clinicians should be especially alert to the bronchospastic effects of adenosine in hospitalized COVID-19 patients. These findings are biologically plausible given what is known about the pathophysiology of COVID-19 and the pharmacology of adenosine, and our study represents the first such report.

Our study has several limitations inherent to a relatively small retrospective analysis. We report on a relatively small number of patients. The higher incidence of prior arrhythmia in the control group suggests that they may have been predisposed to requiring adenosine, but does not influence response to adenosine, and would not be expected to influence likelihood of escalation of care. Similarly, the higher incidence of supplemental O_2_ in the control group was not associated with any differences in pre-adenosine FiO_2_ or O_2_ flow, need for nebulizers, or being on a ventilator. There was a potential for misclassification of true COVID status in patients who were asymptomatic and therefore not re-tested for COVID during their hospitalization. Our definition of “escalation of care” was meant to identify any and all possible adverse consequences of adenosine use, though we recognize that some of the qualifiers (such as increased O_2_ requirements) may not have been clinically important. Another point of discussion is that the two-hour time window following adenosine administration may have been relatively long, since the immediate effects of adenosine last only for a brief duration. However, the adenosine-triggered bronchospastic effects may progress and precipitate clinical deterioration in the ensuing two hours, which explains the reasoning behind the chosen “catchment” window. It is probable that the occurrence of any arrhythmia may be a manifestation of the severity of the underlying illness, and that any escalation of care after adenosine may have been unrelated to the drug. This is consistent with a study from Wuhan, China in which 16.7% of 138 patients admitted for COVID-19 were found to have arrhythmia, with 44.4% of these patients requiring escalation to the intensive care unit [14]. Similarly, it is also possible that any increase in pressor requirement may have been due to the arrhythmia rather than adenosine. Finally, although our matching effort identified a comparable control population, there were some differences, notably a higher incidence of prior arrhythmia, and also greater need for supplemental O_2_ prior to receiving adenosine.

## 5. Conclusions

Our study identified that there is a significant level of hesitation among physicians regarding the use of adenosine for treatment of SVT in the setting of COVID-19 pneumonia, and there are no prior data to address this concern. Within the limitations of a small sample size retrospective review, our data demonstrates that adenosine use in COVID-19 pneumonia patients is not associated with adverse outcomes as compared to non-COVID-19 hospitalized patients with comparable co-morbidity. Adenosine for the treatment of SVT in this setting should not be avoided.

## Figures and Tables

**Figure 1 jcm-12-00969-f001:**
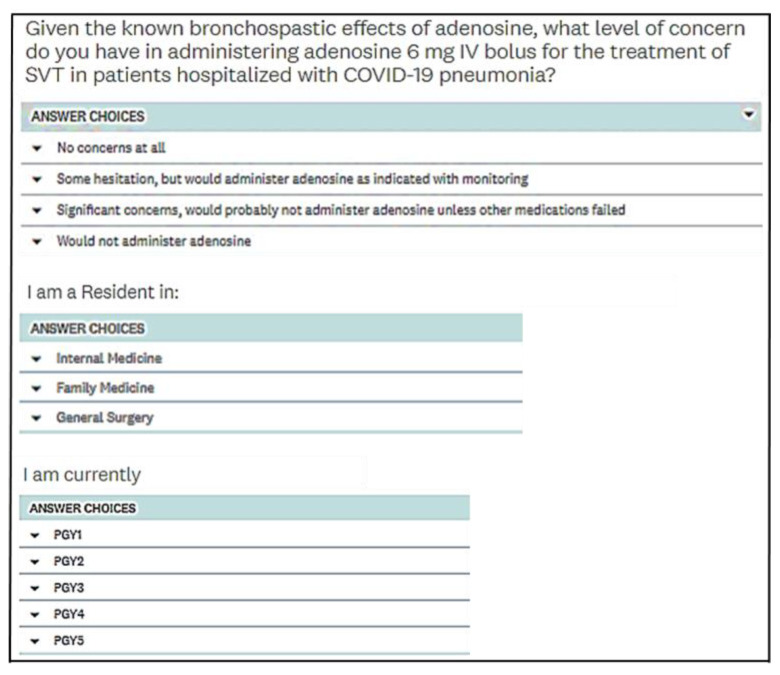
Survey questions sent out to residents.

**Figure 2 jcm-12-00969-f002:**
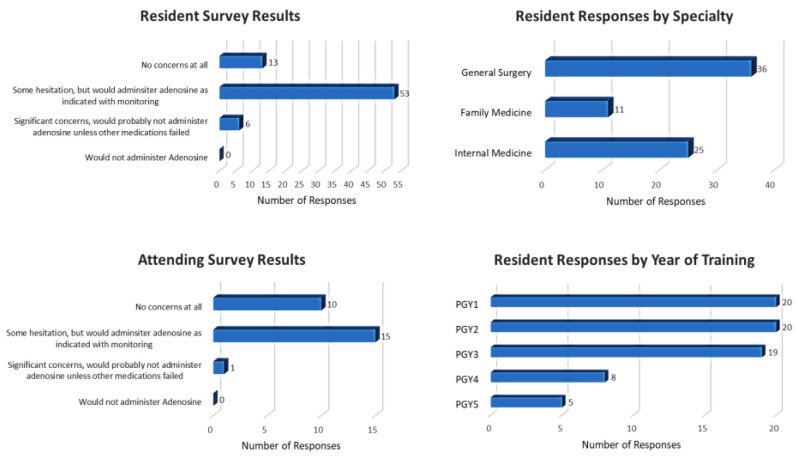
Survey results.

**Table 1 jcm-12-00969-t001:** Baseline characteristics of total population in this study.

Variable	Total Population (N = 84)	Study Population (COVID-19, N = 42)	Control Population (Matched non-COVID-19, N = 42)	*p* Value
Age (years)	65.9	65.7	66.2	0.87
Gender (# male)	64/84 (76.2%)	32	32	1.0
		32	1.0	
		1.0		
History of pulmonary disease	33/84 (39.3%)	18	15	0.50
History of arrhythmia	19/84 (22.6%)	4	15	0.0041
History of any conduction system disease	25/84 (29.8%)	9	16	0.0948
AVNRT was the SVT treated	33/84 (39.4%)	21	11	
Pre-adenosine FiO_2_ (1st admin)	65.5	70.87	58.95	0.10
Pre-adenosine O_2_ flow (1st admin)	14.6	13.42	15.57	0.80
Highest dose of adenosine administered (mg)	8.2	7.86	8.50	0.34
On supplemental O_2_ when adenosine administered	42/84 (50.0%)	12	30	0.0247
Had received nebulizer therapy	28/84 (33.3%)	14	14	1.0
Already on ventilator when adenosine administered	37/84 (44.0%)	18	19	0.83
Already in ICU when adenosine administered	39/84 (46.4%)	21	18	0.51

**Table 2 jcm-12-00969-t002:** Study group. Clinical variables are shown for all study group patients, and also separately for patients who did not and did require escalation of care.

Variable	Study Population (N = 42)	No Escalation After Adenosine (N = 22)	Escalation After Adenosine (N = 20)	*p* Value
Age (years)	65.7	63.2	68.4	0.23
Gender (# male)	32/42 (76.2%)	18	14	0.37
History of pulmonary disease	18/42 (42.9%)	8	10	0.37
History of arrhythmia	4/42 (9.5%)	1	3	0.25
History of any conduction system disease	9/42 (21.4%)	3	6	0.20
AVNRT was the SVT treated	21/42 (50%)	13	8	0.22
Pre-adenosine FiO_2_ (1st admin)	70.87	70.5	71.2	0.95
Pre-adenosine O_2_ flow (1st admin)	13.42	14.8	12.4	0.82
Highest dose of adenosine administered (mg)	7.86	8.0	7.7	0.68
On supplemental O_2_ when adenosine administered	12/42 (28.6%)	5	7	0.38
Had received nebulizer therapy	14 (33.3%)	3	11	0.0045
already on ventilator when Adenosine administered	18 (42.9%)	7	11	0.13
Already in ICU when adenosine administered	21 (50%)	8	13	0.064

**Table 3 jcm-12-00969-t003:** Control group. Clinical variables are shown for all control group patients, and also separately for patients who did not and did require escalation of care.

Variable	Study Population (N = 42)	No Escalation After Adenosine (N = 21)	Escalation After Adenosine (N = 21)	*p* Value
Age (years)	66.2	65.29	67.05	0.68
Gender (# male)	32/42 (76.2%)	17	15	0.47
History of pulmonary disease	15/42 (35.7%)	6	9	0.33
History of arrhythmia	15/42 (35.7%)	7	8	0.75
History of any conduction system disease	16/42 (38.1%)	8	8	0.26
AVNRT was the SVT treated	11/42 (26.2%)	6	5	0.73
Pre-adenosine FiO_2_ (1st admin)	58.95	41.67	66.92	0.0284
Pre-adenosine O_2_ flow (1st admin)	15.57	15.2	15.78	0.97
Highest dose of adenosine administered (mg)	8.50	8.57	8.43	0.88
On supplemental O_2_ when adenosine administered	30/42 (71.4%)	10	20	0.0006
Had received nebulizer therapy	14/42 (33.3%)	4	10	0.0495
Already on ventilator when adenosine administered	19/42 (45.2%)	6	13	0.0495
Already in ICU when adenosine administered	18/42 (42.9%)	7	11	0.21

**Table 4 jcm-12-00969-t004:** Comparison of individual escalation parameters between the study and control groups.

Variable	Total Population (N = 41)	Study Population (COVID-19, N = 20)	Control Population (Matched non-COVID-19, N = 21)	*p* Value
Escalation of O_2_ after adenosine	18/41 (43.9%)	8	10	0.62
Requiring new intubation	2/41 (4.9%)	1	1	0.97
Requiring nebulizer after adenosine	17/41 (41.5%)	12	5	0.0187
Moved to ICU	3/41 (7.3%)	1	2	0.58

## Data Availability

Data sharing not applicable.

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
