# Peer review of "Safety of Adenosine for the Treatment of Supraventricular Tachycardia in Hospitalized Patients with COVID-19 Pneumonia"

_jcm, 2023, doi:10.3390/jcm12030969_

Round 1

Reviewer 1 Report

Well-written manuscript. 

Limitations are properly acknowledged by authors.

Statement on adenosine needs modification according to current evidence. Clinically important bronchoconstriction has been rarely reported in those receiving i.v. adenosine for SVT,1 and this observation is further supported by the large experience obtained when adenosine infusions have been given for cardiac stress testing.2,3Furthermore, despite inhaled adenosine producing bronchoconstriction in people with asthma,4 i.v. administration has no impact on the airways in clinical experimental studies.5 There have been isolated reports of clinically well-documented bronchoconstriction occurring in patients with or without respiratory disease, thus suggesting care is required in patients with asthma.1,6

According to the 2019 ESC GL on SVT, adenosine can be used cautiously in those with asthma, although verapamil may be a more appropriate choice in patients with severe asthma. 

1.         Coli S, et al. Adenosine-induced severe bronchospasm in a patient without pulmonary disease. The American Journal of Emergency Medicine. 2012;30:2082.e2083-2082.e2085. doi: https://doi.org/10.1016/j.ajem.2011.11.005

2.         Layland J, et al. Adenosine: Physiology, Pharmacology, and Clinical Applications. JACC: Cardiovascular Interventions. 2014;7:581-591. doi: https://doi.org/10.1016/j.jcin.2014.02.009

3.         Balan KK, et al. Is the dyspnea during adenosine cardiac stress test caused by bronchospasm? American Heart Journal. 2001;142:142-145. doi: https://doi.org/10.1067/mhj.2001.116070

4.         Cushley MJ, et al. Inhaled adenosine and guanosine on airway resistance in normal and asthmatic subjects. British Journal of Clinical Pharmacology. 2004;58:S751-S755. doi: 10.1111/j.1365-2125.2004.02285.x

5.         Burki NK, et al. The pulmonary effects of intravenous adenosine in asthmatic subjects. Respiratory Research. 2006;7:139-139. doi: 10.1186/1465-9921-7-139

6.         Burkhart KK. Respiratory failure following adenosine administration. The American Journal of Emergency Medicine. 1993;11:249-250. doi: https://doi.org/10.1016/0735-6757(93)90138-2

Author Response

We thank the reviewer for their insight and expertise regarding adenosine. We agree that adenosine induced bronchospasm is rare in non-COVID19 patients, but it is not known if this is also the case in COVID19 patients, given the lack of prior publications. The purpose of our study was to confirm that adenosine is equally safe in COVID19 patients, especially as there are significant concerns about exacerbation of pulmonary symptoms in COVID19 patients amongst all levels of providers. We did not aim to provide a comprehensive review of adenosine as a drug.

Nonetheless, we acknowledge the important comments of Reviewer #1 and have added the following sentences:

In the abstract: Adenosine is commonly used to treat supraventricular tachycardia (SVT), and is generally safe but is rarely associated with bronchospasm. However, there are no data regarding the safety of adenosine in patients with COVID-19 pneumonia, and physicians may hesitate to use adenosine in hospitalized COVID-19 patients.

In the introduction, we have included an additional reference, #7

Reviewer 2 Report

The authors conducted a research study to evaluate safety and perceived safety of administering adenosine for terminating SVT in hospitalized patients with COVID-19. The topic, the research question, and the intended message is novel and important. I thank the authors for their effort and contribution. There are limitations of this study that need to be addressed in a better manner in this manuscript. The low number of participants in the match case-cohort component of the manuscript renders the findings as exploratory. The presentation on the manuscript is also unconventional and appears incomplete. Suggestions:

1) Please expand the methods section to provide more details about how the 2 components in the study were conducted: 

(i) Mention about IRB/ethics committee approval, informed consent

(ii) mention details about how survey was conducted- sampling strategy, response rate, were survey questions validated? 

(iii) In the case-cohort component, how was matching performed to identify controls? the abstract states that the controls & cases were matched on age, sex, and comorbidities, but the methods section states they were matched only on age/sex- please clarify, and if possible provide a table with data on all matching variables to assess whether matching was truly successful.

2) Statistical analysis is not optimal in the case-cohort component of the study. Please calculate odds ratio for odds of escalation in COVID19 patients vs odds of escalation in non-COVID19 pt, with associated p-value & 95% CI. This can be achieved in many ways, for example using online free software like OpenEpi or by using other statistical software.

3) In Table 4 or in results section, please consider providing findings for the composite outcome for escalation (especially because some individual parameters are likely too few resulting in underpowered analysis- i.e. false negatives). 

4) Please try to expand the discussion section with details regarding biologic plausibility of findings and how these findings fit in the existing literature.

5) Please discuss more limitations as is usually done in standard clinical research- potential sources for selection bias, potential unmeasured and unobserved confounding, lack of adequate power for which larger studies are needed, potential misclassification of exposure & outcome (e.g. were all controls selected only if they had a confirmed negative COVID test?) 

Author Response

Response to Reviewer #2: We thank the reviewer for their remarks, and have responded in a point-by-point fashion below.

Suggestions:

1) Please expand the methods section to provide more details about how the 2 components in the study were conducted: 

(i) Mention about IRB/ethics committee approval, informed consent

This is now mentioned at the end of the manuscript after the conclusion (placed at the end by the editors following our initial submission).

(ii) mention details about how survey was conducted- sampling strategy, response rate, were survey questions validated? 

Added how survey was conducted in the methods section. The paragraph has been expanded to read as follows:

“We surveyed resident physicians in the fields of internal medicine, family medicine, and general surgery as well as attending cardiology physicians regarding their level of comfort in administering adenosine to patients with COVID-19 pneumonia. The survey was sent to residents from Internal Medicine, Family Medicine and General Surgery via email with an attached web link. The emails were sent out to the residents by the chief residents of each specialty rather than directly by the study investigators. Reminder emails were sent out once each week for a total of 3 weeks. The survey was also sent to all attendings in the division of cardiology as an email sent directly by the senior author (BP), with 2 reminder emails once a week for 2 weeks. The survey questions are shown in Figure 1. The same survey was also sent to the cardiology attendings, but without the last 2 questions.”

Response rate is shown in results.

(iii) In the case-cohort component, how was matching performed to identify controls? the abstract states that the controls & cases were matched on age, sex, and comorbidities, but the methods section states they were matched only on age/sex- please clarify, and if possible provide a table with data on all matching variables to assess whether matching was truly successful.

Methods and abstract updated to explain matching. The following language has been added:

“The groups were matched for age, sex and comorbidities. Both the study group and control group had 42 patients with a mean age of 66 years old, and a gender split of 10 males and 32 females each. Additionally, the groups were matched for comorbidities including history of pulmonary disease, history of arrhythmia and history of conduction system disease. We obtained relevant medical history, including history of pulmonary disease, history of arrhythmia, and prior use of anti-arrhythmic therapy or rate controlling medications. The highest dosage of adenosine given as well as its effects on cardiac rhythm were documented.”

Table 1 demonstrates equal number of males/females, and average age. We comment on how successful the matching was in the discussion. Equal age and gender but there were more patients with history of arrhythmia in the control group (statistically significant).

2) Statistical analysis is not optimal in the case-cohort component of the study. Please calculate odds ratio for odds of escalation in COVID19 patients vs odds of escalation in non-COVID19 pt, with associated p-value & 95% CI. This can be achieved in many ways, for example using online free software like OpenEpi or by using other statistical software.

We chose not to mention the odds ratios because their numerical closeness to 1.0. At the reviewers’ request, the odds ratios have been added to the abstract and to the results. The following line is now included in the abstract and in the results section:

“The odds ratio of risk of escalation for COVID-19 vs. non-COVID-19 patients was 0.95, 95% CI 0.45-2.01, P=NS).”

3) In Table 4 or in results section, please consider providing findings for the composite outcome for escalation (especially because some individual parameters are likely too few resulting in underpowered analysis- i.e. false negatives). 

The composite outcome is listed in the results, but table 4 just breaks down the individual reasons for escalation.

4) Please try to expand the discussion section with details regarding biologic plausibility of findings and how these findings fit in the existing literature.

Expanded discussion, included biologic plausibility and how it fits into existing literature. The following sentence has been added:

“These findings are biologically plausible given what is known about the pathophysiology of COVID-19 and the pharmacology of adenosine, and our study represents the first such report.”

5) Please discuss more limitations as is usually done in standard clinical research- potential sources for selection bias, potential unmeasured and unobserved confounding, lack of adequate power for which larger studies are needed, potential misclassification of exposure & outcome (e.g. were all controls selected only if they had a confirmed negative COVID test?) 

Revised- discussed lack of adequate power and potential for misclassification. The expanded paragraph regarding limitations now reads as follows:

“Our study has several limitations inherent to a relatively small retrospective analysis. We report on a relatively small number of patients. There was a potential for misclassification of true COVID status in patients who were asymptomatic and therefore not re-tested for COVID during their hospitalization. Our definition of “escalation of care” was meant to identify any and all possible adverse consequences of adenosine use, though we recognize that some of the qualifiers (such as increased O2 requirements) may not have been clinically important. Another point of discussion is that the two hour time window following adenosine administration may have been relatively long, since the immediate effects of adenosine last only for a brief duration. However, the adenosine triggered bronchospastic effects may progress and precipitate clinical deterioration in the ensuing two hours, which explains the reasoning behind the chosen “catchment” window. It is probable that the occurrence of any arrhythmia may be a manifestation of the severity of the underlying illness, and that any escalation of care after adenosine may have been unrelated to the drug. This is consistent with a study from Wuhan, China in which 16.7% of 138 patients admitted for COVID-19 were found to have arrhythmia, with 44.4% of these patients requiring escalation to the intensive care unit.14 Similarly, it is also possible that any increase in pressor requirement may have been due to the arrhythmia rather than adenosine. Finally, although our matching effort identified a comparable control population, there were some differences, notably a higher incidence of prior arrhythmia, and also greater need for supplemental O2 prior to receiving adenosine.”

Reviewer 3 Report

The study (a survey) is potentially interesting and useful for the clinical practice in the setting of COVID-19, but it cannot be considered for publication in the present form. The authors should improve the quality of the manuscript in terms of scientific writing style typo errors and sections’ organization. Please see below my comments:

1.       The main weakness of the manuscript is the extremely short discussion, which is mainly a list of study limitations, and which undoubtedly reduce the quality of the study. Authors should consider improving this section by discussing more in detail their main findings. Listed statements should not be present in the discussion

2.       At the end of the introduction, subhead title should be removed, and the objectives should not be listed

3.       In the authors name line, annotations such as MD and MPH should be upper score

4.       Please provides study/control groups numerosity in the abstract

5.       Please includes ACE2, COVID-19 and SARS-CoV-2 (please check also for other acronyms throughout the text) in their complete name when firstly mentioned

6.       Adenosine mediates its activity upon binding to its receptors A1 ( PMID: 23291003), A2 (PMID: 1513184) and A3 (PMID: 34750517). For completeness, this information and these references should be included as a background

7.       Reference style should be carefully checked and uniformed. Several inaccuracies are present throughout the text. In general I also suggest including more supporting reference in the introduction.

8.       Please includes at least sample size, mean ages and gender ratio in the methods section

9.       2.2 and 2.3 section can be merged

10.   No supporting reference were included in the statistical analysis section

11.   For a better reading, suggest switching the percentage and the ratio of the results. For instance, “59/72 or 82%” should be “82% (59/72). “20/84 (23.8%)” should be “23.8 (20/84)” etc..  

12.   Significant p values should not be in red

Author Response

Response to Reviewer #3: We thank the reviewer for their remarks, and have responded in a point-by-point fashion below.

  1. The main weakness of the manuscript is the extremely short discussion, which is mainly a list of study limitations, and which undoubtedly reduce the quality of the study. Authors should consider improving this section by discussing more in detail their main findings. Listed statements should not be present in the discussion

Fixed discussion layout. Removed listed statements. Expanded discussion for further relevant points.

  1. At the end of the introduction, subhead title should be removed, and the objectives should not be listed.

Removed

  1. In the authors name line, annotations such as MD and MPH should be upper score

Now upper score

  1. Please provides study/control groups numerosity in the abstract

Numbers now included in abstract

  1. Please includes ACE2, COVID-19 and SARS-CoV-2 (please check also for other acronyms throughout the text) in their complete name when firstly mentioned

Full name provided for all abbreviations when first mentioned

  1. Adenosine mediates its activity upon binding to its receptors A1 ( PMID: 23291003), A2 (PMID: 1513184) and A3 (PMID: 34750517). For completeness, this information and these references should be included as a background

We agree that adenosine has well-described cellular and molecular actions, and respect the level of comprehension that the reviewer clearly has. However, the purpose of our clinical study was to demonstrate that adenosine is as safe in COVID19 patients as in non-COVID19 patients. We did not aim to provide a comprehensive review of adenosine as a drug. Therefore, we humbly disagree with this requirement to describe the A1, A2 and A3 receptor actions of adenosine.

  1. Reference style should be carefully checked and uniformed. Several inaccuracies are present throughout the text. In general I also suggest including more supporting reference in the introduction.

Edited the inaccuracies, ensured the style is uniform, and added references.

  1. Please includes at least sample size, mean ages and gender ratio in the methods section

Sample size, mean age and gender are now included in methods

  1. 2.2 and 2.3 section can be merged

Merged

  1. No supporting reference were included in the statistical analysis section

The statistical methods used are standard (student’s t test, Fisher’s exact test, and odds ratios). The authors respectfully believe that these do not need referencing.

  1. For a better reading, suggest switching the percentage and the ratio of the results. For instance, “59/72 or 82%” should be “82% (59/72). “20/84 (23.8%)” should be “23.8 (20/84)” etc..  

Switched the percentage and ratio as suggested.

  1. Significant p values should not be in red

Significant p values are no longer in red.

Round 2

Reviewer 2 Report

Thank you for submitting a revised manuscript. The manuscript looks good. Just a minor suggestion for your consideration:

In the abstract, the conclusion reads, "The data demonstrate that adenosine use is comparably safe in COVID-19 and matched non-COVID-19 hospitalized patients. Adenosine for treatment of SVT in this setting should not be avoided." However, the study was underpowered and absence of evidence is not the same as evidence of absence. A more accurate conclusion could be: " In this study, there was no evidence of increased harm from administering adenosine to patients with SVT and COVID-19. This finding needs to be confirmed in larger studies. Based on current evidence, adenosine for treatment of SVT in this setting should not be avoided."

Author Response

Thank you for taking the time to write your review and to provide feedback. Our team agrees with your suggestion, and the abstract conclusion has been modified to reflect this. See below abstract, as well as the final manuscript edit attached. 

"Abstract: Coronavirus disease 2019 (COVID-19) is associated with pulmonary involvement and cardiac arrhythmias, including supraventricular tachycardia (SVT). Adenosine is commonly used to treat SVT and is generally safe, but is rarely associated with bronchospasm. There are no data regarding the safety of adenosine in patients with COVID-19 pneumonia and physicians may hesitate to use it in such patients. We surveyed resident physicians and cardiology attendings regarding their level of comfort in administering adenosine to hospitalized COVID-19 patients. We compared a study group of 42 COVID-19 hospitalized patients who received adenosine for SVT to a matched (for age, sex, and co-morbidities) control group of 42 non-COVID-19 hospitalized patients during the same period, all of whom received IV adenosine for SVT. Escalation of care following intravenous adenosine administration was defined as increased/new pressor requirement, need for higher O2 flow rates, need for endotracheal intubation, new nebulizer therapy, or transfer to intensive care unit within 2 hours of adenosine administration. Survey results showed that 82% (59/72) of residents and 62% (16/26) of cardiologists expressed hesitation/significant concerns regarding administering adenosine in hospitalized COVID-19 patients. Adenosine use was associated with escalation of care in 47.6% (20/42) COVID-19 as compared to 50% (21/42) non-COVID-19 patients (odds ratio 0.95, 95% CI 0.45-2.01, P=NS). Escalation of care was more likely in patients who were on higher FiO2, on prior nebulizer therapy, required supplemental oxygen, or were already on a ventilator. In conclusion, we identified significant hesitation among physicians regarding use of adenosine for SVT in hospitalized COVID-19 patients. In this study, there was no evidence of increased harm from administering adenosine to patients with SVT and COVID-19. This finding needs to be confirmed in larger studies. Based on current evidence, adenosine for treatment of SVT in this setting should not be avoided."
